# E-Cadherin/HMR-1 Membrane Enrichment Is Polarized by WAVE-Dependent Branched Actin

**DOI:** 10.3390/jdb9020019

**Published:** 2021-05-07

**Authors:** Luigy Cordova-Burgos, Falshruti B. Patel, Martha C. Soto

**Affiliations:** Department of Pathology and Laboratory Medicine, Rutgers—RWJMS, Piscataway, NJ 08854, USA; lc1022@rwjms.rutgers.edu (L.C.-B.); falubp24@live.com (F.B.P.)

**Keywords:** epithelial polarity, branched actin, Cadherin, apical junctions

## Abstract

Polarized epithelial cells adhere to each other at apical junctions that connect to the apical F-actin belt. Regulated remodeling of apical junctions supports morphogenesis, while dysregulated remodeling promotes diseases such as cancer. We have documented that branched actin regulator, WAVE, and apical junction protein, Cadherin, assemble together in developing *C. elegans* embryonic junctions. If WAVE is missing in embryonic epithelia, too much Cadherin assembles at apical membranes, and yet apical F-actin is reduced, suggesting the excess Cadherin is not fully functional. We proposed that WAVE supports apical junctions by regulating the dynamic accumulation of Cadherin at membranes. To test this model, here we examine if WAVE is required for Cadherin membrane enrichment and apical–basal polarity in a maturing epithelium, the post-embryonic *C. elegans* intestine. We find that larval and adult intestines have distinct apicobasal populations of Cadherin, each with distinct dependence on WAVE branched actin. In vivo imaging shows that loss of WAVE components alters post-embryonic E-cadherin membrane enrichment, especially at apicolateral regions, and alters the lateral membrane. Analysis of a biosensor for PI(4,5)P2 suggests loss of WAVE or Cadherin alters the polarity of the epithelial membrane. EM (electron microscopy) illustrates lateral membrane changes including separations. These findings have implications for understanding how mutations in WAVE and Cadherin may alter cell polarity.

## 1. Introduction

Adherens junctions establish the apical domain of epithelia through the connection of the Cadherin complex to filamentous actin (F-actin). Recent studies have shown that the connection between Cadherin-based apical junctions and F-actin is surprisingly dynamic. Our in vivo studies in *C. elegans* have shown that WAVE-dependent branched F-actin contributes to the apical junctions, during their establishment, and during their maintenance [1,2]. To address precisely how branched actin promotes adhesion and polarity through Cadherin will require further development of in vivo models where endogenous Cadherin can be visualized and where temporal depletion of branched actin regulators is possible.

We have used the *C. elegans* embryo as a model for understanding the polarized forces of morphogenesis. Genetic screens led us to focus on one major driver of embryonic morphogenesis, in *C. elegans* and other organisms, the WAVE Complex. WAVE is a nucleation promoting factor (NPF) that creates branched F-actin by activating the Arp2/3 complex. WAVE is the major NPF that turns on branched actin through Arp2/3 during *C. elegans* and Drosophila embryonic morphogenesis, despite that fact that other NPFs act in embryos, including WASP [1,3,4]. WAVE is itself under tight regulation, and must be recruited to membranes to act. We have identified signals at the membrane that activate the GTPase Rac1/CED-10, which in turn binds and activates WAVE, so it can promote branched actin through Arp2/3 [5].

The *C. elegans* apical junction (CeAJ) is composed of at least two functionally distinct complexes, yet has a single electron dense junctional complex, compared to two in vertebrates and *Drosophila* (reviewed in [6]). The more apical Cadherin Catenin Complex (CCC) contains *C. elegans* homologs of E-Cadherin/HMR-1, beta-catenin/HMP-2 and alpha-catenin/HMP-1 [7]. The more basally localized DAC junction contains homologs of Discs Large/DLG-1 and AJM-1 [8,9,10]. *C. elegans* has a homolog of the tight junction component ZO-1, ZOO-1, that is recruited to the Cadherin junction by E-Cadherin/HMR-1, but it does not appear to be required for paracellular gate function [11,12]. The apical-most complex in *C. elegans* is the Cadherin complex, not a separate Tight Junction. Therefore, to understand the apical junction we focus here on *C. elegans* E-Cadherin, which we will refer to hereafter as Cadherin/HMR-1.

We previously showed that in the embryonic intestine, WAVE complex components are enriched just apical to the CeAJ, suggesting WAVE is at the correct place to support junctions [1,13]. Paradoxically, in embryos, loss of WAVE leads to excess Cadherin accumulation, coupled with decreased apical F-actin [2]. To understand these complex results, it helps to consider that branched actin may play multiple roles in regulating epithelial Cadherin. Actomyosin sets up the tension that maintains apical Cadherin [14,15,16]. While actomyosin appears to mainly include linear actin, in human umbilical vein endothelial cells (HUVECs), platinum replica electron microscopy (PREM) illustrated that actomyosin linear fibers are connected to VE-Cadherin through branched actin [17]. Thus, one function for branched actin at apical junctions may be to link the Cadherin complex to actomyosin in the apical actin ring that supports polarized epithelia.

A second proposed role for branched actin at junctions to push adult epithelial membranes together, for example when individual cells join an epithelium, and to repair breaks in adhesion [18,19,20]. Branched actin may help maintain epithelial Cadherin in adult tissues that need to repair damage. For example, branched-actin dependent microspikes were reported in MDCK and Caco-2 cells, and proposed to permit repair of breaks in lateral adhesion [21]. Our studies provide evidence that WAVE and Cadherin maintain apical junctions in an organismal setting. In an adult epithelium, the *C. elegans* intestine, we used the F-actin dye phalloidin to show that regulators of branched actin (including WAVE), of linear actin (including the formin Diaphonous/CYK-1) and the Cadherin complex support the apically enriched F-actin of the adult intestine, while their loss reduced apical F-actin and resulted in an expanded apical lumen [1]. However, effects at lateral membranes were not examined.

Cadherin is known to regulate adhesion, yet in *C. elegans*, embryonic loss of Cadherin does not eliminate overall adhesion, likely due to redundancy with other adhesive molecules [7]. However, embryonic Cadherin contributes to apicobasal polarity: it is needed for apical F-actin enrichment in the developing intestine [1,2] and PAR-dependent apicobasal polarity in blastomeres [22]. We have found a shared role for Cadherin and the WAVE branched actin regulator in establishing embryonic apical junctions, and the proteins promote each other’s embryonic apical enrichment. In adults, we showed using RNAi depletion and electron microscopy (EM) that Cadherin and WAVE help maintain apical junctions [2]. However, the subcellular distribution of adult Cadherin in a tissue that permits dissection of apicobasal distribution and regulation has not been reported.

Apical–basal polarity in the epithelia is supported by polarized phospholipids, major components of biological membranes, which are themselves polarized by the distribution of polarity regulators. For example, some phosphoinositides (PI) are enriched at apical regions, where they help recruit membrane associated proteins, and help set up subcellular trafficking compartments (reviewed in Krahn 2020). Thus, if Cadherin and WAVE are mutually setting up polarity, changes in Cadherin or WAVE membrane enrichment may be reflected in changes in the distribution of phosphoinositides, like PI(4,5)P2, which is polarized to apical membranes, where it promotes polarized vesicle transport (Thapa and Anderson 2012). 

To determine if WAVE-dependent membrane enrichment of Cadherin documented in embryos also occurs in post-embryonic larval and adult epithelia, and to explore possible mechanisms that contribute to this regulation, here we analyzed the membrane enrichment of Cadherin/HMR-1 that was fused to a fluorescent protein at its naturally occurring chromosomal location using CRISPR technology [23]. We show that the distribution of Cadherin/HMR-1 in larval and adult intestines is highly polarized, and depends on WAVE. We generate a new transgenic strain to monitor apical F-actin in the post-embryonic intestine, to test for a shared role for WAVE and Cadherin in maintaining this epithelial apical F-actin. Analysis of the lateral junction suggests WAVE is particularly relevant to supporting the apicolateral pool of Cadherin. Live imaging and EM analysis of the larger postembryonic intestinal cells, relative to embryonic cells, reveals altered lateral membranes, and reduced recruitment of apical phosphoinositides in epithelia depleted of Cadherin or WAVE components. The data shown here support our hypothesis that maintenance of correct Cadherin localization is dependent on WAVE-dependent branched actin and suggest future directions to address the mechanism.

## 2. Materials and Methods

*C. elegans* strains built for this paper: To build intestinal F-actin strain *OX966 Pglo-1::Lifeact::TAG::RFP; pRF4 rol-6(su1006)*, we cloned Lifeact::TAG::RFP from clone pDS355 (gift from Daniel Shaye) behind the intestinal *glo-1* promoter (gift from Greg Hermann). The plasmid was verified by sequencing, and injected according to Mello et al., 1991. After UV-TMP integration we selected a strain with consistent rolling and Lifeact signal in the intestine. 

The following strains which contain fluorescent proteins fused at naturally occurring chromosomal locations using CRISPR technology: *LP238 hmr-1::mKate2*, and *LP431 gfp::3xFLAG-gex-3* [23] as well as *RT1120 Pvha-6::PH::gfp* (Barth Grant, Rutgers University) were combined to create doubly marked strains: *OX970 Pvha-6::PH::gfp; hmr-1::mKate2, OX974 hmr-1::mKate2; gfp::3xFLAG-gex-3* and *OX975 Pglo-1::Lifeact::TAG::RFP; Pvha-6::PH::gfp*.

*RNAi experiments*: All RNAi bacterial strains used in this study were administered by the feeding protocol as in [5]. RNAi feeding experiments were performed at 23 °C unless otherwise mentioned. Worms were synchronized and transferred onto seeded plates containing RNAi-expressing bacteria, or “Control” bacteria carrying the RNAi vector L4440 with no gene inserted. Effectiveness of the RNAi was monitored after 2 days using two methods, and worms were imaged on day 3 if both methods showed an effect. (1) We counted the percent dead embryos, which after two days is expected at >90% for *gex-3, gex-2 or arp-2,* and at least 40% for *hmr-1*, 50% for *hmp-2* and 20% for *hmp-1*. (2) We monitored post-embryonic silencing of a gfp-tagged strain in the intestine, such as *gfp::gex-3* or *hmr-1::gfp* by the corresponding RNAi. RNAi experiments with the expected embryonic lethality, and knock-down of the gfp signal of at least 75% for *gex-3* or *hmr-1*, 24% for *hmp-2* and 5% for *hmp-1* were used. RNAi controls are included in Appendix A.

*Live Imaging*: Imaging was performed in a temperature-controlled room set to 23 °C on a Laser Spinning Disk Confocal Microscope with a Yokogawa scan head, on a Zeiss AxioImager Z1 Microscope using the Plan-Apo 63X/1.4NA or Plan-Apo 40X/1.3NA oil lenses. Images were captured on a Hamatsu CMOS Camera using MetaMorph software, and analyzed using ImageJ. Controls and mutants were imaged within 3 days of each other with the same imaging conditions. All measurements were performed on raw data using ImageJ. For fluorescent measurements, background intensity was subtracted by using a box or line of the same size and measuring average intensity in the same focal plane, near the animal. 

*Microscopy of adults:* Young adult stage animals (one day after L4 stage) and L4 larvae were placed on agar pads in M9 solution and immobilized using 10 μL of Levamizole(10 μm) salts and covered with 1.5 um coverslips. Images were taken within 15 min of making the pads. Imaging was performed on the Zeiss AxioImager Z1 with a Yokogawa CSUX1-5000 spinning disc, using the Plan Apo 63X/1.4NA Oil lens. Images were analyzed and measured using ImageJ.

*Quantitation of immunofluorescence:* Quantitation of live fluorescence was performed using the line selection and the dynamic profile function of ImageJ to measure fluorescence along lines of equal length along, for example, the apical intestine or lateral regions of the intestinal cells, as indicated in Figure 1, Figure 2 and Figure 3. For all experiments shown, the images were captured at the same exposure settings for controls and mutants. All quantitation was performed on the raw images. The figure legends indicate when images were enhanced for contrast, and the same enhancement was applied to a mosaic of the related images for that experiment.

*Electron Microscopy:* was performed as described in [1] but adapted for adults. 

Synchronized L1s were plated on control food or on RNAi food for 2d for *arp-2*, 3d for others. Samples were washed in M9 Buffer, anesthetized with 10 mM levamisole, then fixed for 3 h–4 h in fresh 4% paraformaldehyde (15710; Electron Microscopy Sciences, Hatfield, PA, USA) and 2% glutaraldehyde (16220; Electron Microscopy Sciences) in 0.1 M HEPES pH 7.2, after cutting near the pharynx to preserve the intestine. After fixation worms were washed three times for 3 min in M9 or 0.2 M HEPES, aligned with an eye lash with heads in one direction, and gently covered in 2% agarose heated to 55 degrees. M9 was replaced with 1% OsO4 for a 1 h–2 h incubation. Samples were dehydrated on a nutator at room temperature for 10 min sequentially with 50% ethanol, 70% ethanol, 95% ethanol, and 100% ethanol and twice with 100% acetone. The samples were then incubated in a series of graded Epon/acetone solutions. Epon/acetone (1:1) solution was used at room temperature for 2 h, followed by (3:1) Epon/acetone incubation overnight at room temperature on a nutator. The solution was replaced with 100% resin and microwaved in a Pelco microwave (Ted Pella, Redding, CA, USA) at 250 W with the vacuum on for 3 min. The microwave cycle was repeated with new 100% resin, and the vacuum was left on for an additional 30 min. The agarose block was cut into sections, and resin was then replaced with clean 100% resin, spun, and baked at 70 °C overnight. Blocks were trimmed, and 80-nm to 90-nm thin sections were cut using a Diatome diamond knife. The sections were picked up on 200-mesh, thin-bar copper grids and stained with uranyl acetate and lead citrate. Samples were imaged using an Advanced Microscopy Techniques (Woburn, MA, USA) camera on a Philips (Briarcliff Manor, NY, USA) CM12 electron microscope.

*Statistical Analysis:* For grouped data statistical significance was established by performing a one-way Analysis of Variance (ANOVA) followed by a Sidak’s multiple comparisons test, for Figure 1C, and Dunnett multiple comparison post-test for other figures. For ungrouped data an unpaired t-test, the unequal variance (Welch) t test was used. Error bars show 95% confidence intervals. Asterisks (*) denote *p* values * = *p* < *0*.05, ** = *p* < 0.001, *** = *p* < 0.0001, *** = *p* < 0.00001. All statistical analysis was performed using GraphPad Prism 8.

## 3. Results

### 3.1. Cadherin Is Highly Polarized in the Intestinal Epithelium 

We set out to characterize the apical/basal distribution of Cadherin/HMR-1 in the *C. elegans* post-embryonic intestine, an epithelium of just 20 cells that must last the entire life of the animal. Previous studies of Cadherin/HMR-1 in adult *C. elegans* relied on integrated multicopy arrays [24], which may not reflect endogenous Cadherin enrichment. Perhaps for this reason, using these transgenes, a previous study suggested Cadherin/HMR-1 was apically enriched in adult epidermal cells, but failed to detect any apical enrichment of Cadherin/HMR-1 in the adult intestine [25]. 

We therefore analyzed the distribution of endogenous Cadherin/HMR-1 in L4 larvae and adults using new CRISPR tagged strains [23] focusing on the intestine (Figure 1). The *C. elegans* intestine is a tube that runs for most of the length of the animal. As part of the alimentary system it attaches to the pharynx and valve cells at its anterior, and to the anal opening at its posterior. The entire intestine is made up of only 20 cells, with 4 cells at the anterior segment and 2 cells per segment for the remaining 8 segments. Adult worms are imaged on agar pads, which results in the animals usually being imaged on their sides. Imaging through the transparent worm, from top to bottom first shows a basal view of the intestine (basal focus, Figure 1A,B), where one sees, at the center, the basal junction between two cells. Focusing at the middle of the intestine, (apical focus) allows imaging of the central apical lumen, basal regions at top and bottom, and linking the two, the lateral junction between two intestinal cells (Figure 1A,B). We chose this apical focus for most of the imaging reported here to display apical, basal and the lateral regions between the intestinal rings in a single Z slice. We found that by the L4 larval stage Cadherin/HMR-1 signal is strong in the intestine. Since the pattern in L4s is consistent with the adult pattern, we performed most the comparisons shown here at the L4 stage, which has lower intestinal autofluorescence than adults. To minimize imaging interference from the germline, we focused on rings 2 and 3 of the 9-ring intestinal tube, just posterior to the pharynx. The L4 stage was confirmed by imaging the developing vulva (“Christmas Tree” stage, which marks mid-L4), the position of the gonad arms, and finally the status of the epidermal alae, which are unfused in L4 and fuse in young adults.

We measured the enrichment of Cadherin along the apicolateral, midlateral, and basolateral subregions of the lateral junction, focusing on the junction between intestinal rings 2 and 3 (Int 2/3, Figure 1B). Similarly to Cadherin distribution in other epithelial systems, the highest enrichment of Cadherin/HMR-1 was measured at the apicolateral region. Therefore, as suggested by EM images of the adult intestine, and in contrast to what was shown using multicopy arrays, endogenously tagged Cadherin/HMR-1 is apically enriched within the lateral membrane (apicolaterally) in larval and adult intestines. By comparison, there is low Cadherin/HMR-1 enrichment along the lumen, while a surprising amount is enriched at basal membranes (Figure 1B). As expected, Cadherin/HMR-1 is also enriched in the ventral and dorsal nerve cords, outside the intestine. 

### 3.2. Cadherin Apicobasal Distribution Depends on WAVE

We had reported that in embryonic intestine there is a significant increase in apical Cadherin/HMR-1 levels when we reduced WAVE components. However, in the tiny embryonic intestine, we could not visualize lateral Cadherin/HMR-1, only the Cadherin/HMR-1 that accumulates along the lumen [2]. When we removed WAVE components in L4 larvae and adults, we detected significantly reduced apicolateral Cadherin/HMR-1, and relatively unchanged levels at midlateral and basolateral regions of the intestine (Figure 1B,C). Therefore, postembryonically WAVE regulates Cadherin/HMR-1 accumulation particularly at apicolateral membranes in the intestine, the region that by EM contains the single electron dense apical junction. 

### 3.3. WAVE and Cadherin Complexes Regulate Apical F-Actin in the Intestinal Epithelium

Since loss of WAVE had the largest effects on apicolateral populations of Cadherin, we wanted to next test if WAVE and Cadherin were contributing to apical lumen F-actin accumulation in vivo, as we saw with fixed adults and phalloidin staining. We therefore developed a new reagent to detect F-actin in adult intestines using the *glo-1* promoter [26], driving Lifeact fused to TAG-RFP (gift of Daniel Shaye), marked with the *rol-6* dominant Roller mutation [27]. We made integrated lines of this new strain that allowed us to monitor F-actin accumulation in the developing embryonic intestine and all through larval and adult stages (Figure 2A,B). As we have seen with other markers for endogenous F-actin, *Pglo-1::Lifeact::TAG-RFP* becomes enriched apically as the embryonic intestine becomes polarized. As the embryos develop, the signal becomes more enriched apically. In L4 larvae and adults, most of the expression is found at apical regions surrounding the lumen, while lower enrichment is seen at lateral regions (Figure 2B).

We tested if apical F-actin identified by *Pglo-1::Lifeact::TAG-RFP* was affected by loss of the WAVE complex, or the Cadherin complex. Removing the WAVE complex branched actin regulators *gex-3* or *gex-2* via RNAi, significantly reduced apical F-actin (Figure 2C). Loss of components of the two main adherens junction complexes, Cadherin/*hmr-1*, beta-catenin/*hmp-2* or a-catenin/*hmp-1*, or *dlg-1*, via RNAi, significantly reduced F-actin levels, just as we previously showed for phalloidin (Figure 2B,C; [1]). This result supports that the Cadherin and WAVE complexes are required to establish and maintain robust enrichment of apical lumen F-actin in the larval and adult intestine. Thus the apicolateral, WAVE-dependent population of Cadherin (Figure 1B,C), appears to support apical lumen F-actin in this mature epithelium.

While the *Pglo-1::Lifeact::TAG-RFP* strain is highly enriched along the apical lumen of the intestine, we noted a lower enrichment at the lateral membrane of the intestine in control embryos. To test if this lateral F-actin depends on Cadherin and WAVE components, we compared controls to animals with RNAi depletion of Cadherin*/hmr-1* or WAVE component *gex-2*. Controls show a thin, uniform accumulation of lateral F-actin from apical to basal regions of the lateral membrane (Figure 2D). Animals depleted of *hmr-1* or *hmp-1* often showed non-uniform clumps or swellings of F-actin in parts of the lateral membrane. Viewing these same Z slices with DIC optics suggests these may be regions of membrane alteration (Figure 2D). Control animals never show these swellings (*n* > 25) while mutants showed swellings and membrane bubbles (apparent vesicles) in 9/23 *hmr-1* and 3/8 *hmp-1* RNAi animals. *gex-2* RNAi animals show extremely low levels of lateral F-actin (Figure 2D). This result shows changes in Cadherin/HMR-1 accumulation at the apicolateral membrane (Figure 1) correlate with changes in more basal regions of the lateral membrane (Figure 2). 

### 3.4. WAVE and Cadherin Colocalize in Intestinal Epithelia

If WAVE and Cadherin collaborate to polarize F-actin in the larval and adult intestine, it is important to view if they colocalize subcellularly in this epithelium. Imaging the embryonic intestine in fixed embryos, we previously showed that WAVE complex components are most enriched just apical to the Cadherin junction [1]. In the embryonic epidermis, we showed the WAVE complex is most enriched just basal to the Cadherin and DLG-1/AJM-1 complexes [13]. We therefore wanted to examine the relative enrichment of WAVE Complex and Cadherin Complex components in the post-embryonic intestine using in vivo imaging and endogenously tagged strains. L4 stage larvae were imaged in a strain with two endogenously tagged WAVE and Cadherin components, *hmr-1::mKate2; gfp::gex-3* (*OX974,* see Methods). In these animals, the signal from Cadherin/HMR-1 and WAVE component GEX-3 clearly overlapped at the apicolateral membrane (Figure 2E). Therefore, WAVE is enriched subcellularly in the region where it seems to most affect Cadherin enrichment, and the WAVE and Cadherin show similarly polarized apicobasal enrichment.

### 3.5. Cadherin Lateral Distribution and Membrane Polarity Depends on WAVE

Loss of WAVE appears to disrupt not just the apicolateral enrichment of Cadherin/HMR-1 in the adult intestine, but to alter the morphology of the lateral membrane (Figure 1 and Figure 2). To better analyze how branched actin regulates in the membrane distribution of Cadherin, we made use of a biosensor for membrane lipid PI(4,5)P2 (PIP2). PIP2 is apically enriched in epithelia, and loss of PIP2 alters polarity, since polarity determinants and phosphoinositides reinforce membrane polarity (reviewed in [28,29]). The *Pvha-6::PH::gfp* strain expresses the PIP2 binding PH domain of rat PLC-delta under the *vha-6* intestinal promoter [30]. We therefore imaged worms carrying *hmr-1::mKate2*, or *Pvha-6::PH::gfp*, or both. *Pvha-6::PH::gfp* is highly enriched at the apical membrane lining the lumen of the intestine, and much lower enrichment is detected at basolateral membranes (Figure 3A). We measured the average signal at apical and midlateral membranes, in controls and in animals depleted of *gex-3* or *hmr-1* with 2 day RNAi feeding treatment. In controls, there is high apical enrichment, with a ratio of apical to lateral of 2.5. By contrast, the treated animals showed significantly reduced apical signal, a drop of 25% for *gex-3* and 46% for *hmr-1* (Figure 3B). While the enrichment at all membranes was reduced, the largest change was at apical membranes, which suggested a change in membrane polarity.

We further examined animals carrying both *hmr-1::mKate2*, in magenta, and *Pvha-6::PH::gfp*, in green, to ask if changes at lateral regions correspond to changes in Cadherin enrichment (Figure 3C). In control animals the lateral membranes marked with *hmr-1::mKate2* appear as uniform, thin regions. When *gex-3* was reduced, some lateral regions became more diffuse and wider (Figure 1 and Figure 3). Measuring the width of the membrane showed an average of 0.41 microns for controls and 0.55 microns for *gex-3* RNAi animals (Figure 3D). While apicolateral and basolateral regions remain attached, and co-express HMR-1::mKate2 and PH::GFP, changes are visible at midlateral regions, including reduced accumulation of both HMR-1::mKate2 and PH::GFP (Figure 3C), and some *hmr-1 RNAi* animals appeared to have lateral membrane separations Figure 3C). 

To examine how changes in lateral F-actin corresponded to changes at the lateral membrane, we next crossed the new *Pglo-1::Liifact::TAG-RFP* strain into *Pvha-6::PH::gfp* (Figure 3E). While lateral F-actin enrichment illustrated by *Pglo-1::Lifeact::RFP* is thin and uniform in controls, loss of *gex-3* by RNAi resulted in altered F-actin that corresponded to membrane expansions and deformations. Loss of Cadherin/*hmr-1* via RNAi also resulted in membrane expansions and deformation, and shifts in F-actin enrichment at subsets of the lateral membranes. In some animals depleted of *hmr-1* or *hmp-1*, there were areas of reduced F-actin that corresponded with large lateral membrane extensions surrounded by PH::GFP, and devoid of F-actin (Figure 3E, panels with two white arrows). This raised the possibility that these F-actin and Cadherin changes were affecting the lateral membrane directly. 

### 3.6. Changes in the Lateral Membrane in Animals Depleted of WAVE or Cadherin Components Are Visible by EM

To view lateral membrane changes at the highest resolution, we reviewed electron micrographs (EM) of adult intestines with a focus on lateral membranes. As we had previously reported, loss of WAVE complex components, or Cadherin components, led to shorter and less electron dense junctions (Figure 4; ref. [2]). Control animals showed uniformly spaced lateral membranes (dotted white lines). Reducing the WAVE complex via *gex-3* RNAi resulted in some animals with alterations of the lateral membrane including regions of mild lateral expansions. Loss of alpha catenin/*hmp-1* via RNAi similarly showed regions of increased lateral distances. In animals depleted of the Arp2/3 complex via *arp-2* RNAi, even larger regions of membrane separation were seen (Figure 4).

In summary, either by live imaging, or by examining fixed animals viewed by EM, we documented changes in lateral membrane of the intestine when Cadherin/HMR-1 or WAVE components including *gex-2* and *gex-3* were depleted via RNAi feeding. We did not see complete loss of adhesion, and instead noted small gaps appearing, visible by EM (Figure 4) and sometimes by DIC (Figure 2 and Figure 3).

## 4. Discussion

In vivo studies and high-resolution imaging studies support that even after epithelia form and polarize, they continue to require support from branched actin. This study sets up a system in which to test models for the mechanisms that promote epithelial polarity through the force of branched actin.

We first addressed a puzzle in our field. Previous reports suggested Cadherin/HMR-1 was not polarized in the adult intestinal epithelium. This result seemed incongruous to us, since it was clear Cadherin was required for apical F-actin enrichment in this tissue, as shown by phalloidin staining that was significantly diminished when Cadherin was depleted [1]. 

When CRISPR tagged Cadherin/HMR-1 became available [23], we decided to revisit the question of how Cadherin is distributed in this mature epithelium. The images and measurements in Figure 1 show that Cadherin/HMR-1 is indeed polarized from apical to basal regions. This polarized enrichment depends on WAVE (Figure 1). To determine if WAVE and Cadherin/HMR-1 mutually support apical F-actin, we built a new strain to image apical F-actin in the adult intestine. This strain, *Pglo-1::Lifeact::TAG-RFP*, shows that apical F-actin enrichment depends on WAVE complex-dependent branched actin, and Cadherin. 

Imaging Cadherin/HMR-1 also revealed a substantial level of Cadherin/HMR-1 along basal membranes (Figure 1). The basal intestine faces the pseudocoelomic space but is not thought to bind to other cells. It will be interesting to investigate if this pool of Cadherin is high due to the delivery of Cadherin to the basal regions for later transport to the apicolateral membranes, as is proposed in the Drosophila intestine [31]. In this case, this pool of Cadherin might be an inactive reservoir formed by internal trafficking events. Alternatively, the basal intestine may experience shear forces or other mechanical stimuli that would require the establishment of a basal pool of Cadherin. In this latter case, the basal pool may be functional, and perhaps under external regulation. However, since the basal cells are not thought to adhere to other cells, we favor the idea that this is a reservoir of Cadherin that can be delivered to the apicolateral region as needed. 

We further investigated if the effects of reducing branched actin created polarity defects, or adhesion defects for the Cadherin. At first glance, it did not seem that loss of branched actin by reducing WAVE components caused loss of adhesion, since, for example, we did not see large separations of the basolateral membrane. However, the changes in the Cadherin enrichment, including the diffuse signal at mid-lateral regions (for example, Figure 3A,D) led us to look more carefully at the lateral regions. Using *Pglo-1::Lifeact-TAG-RFP* to image intestinal F-actin, we saw F-actin accumulations in animals depleted of Cadherin or WAVE components that were much lower than is seen in controls (Figure 2). Since Cadherin is expected to connect to the apical F-actin belt of this epithelium, the reduced lumenal F-actin may reflect a drop in the apical/basal polarity of the cells, and/or changes in the apical lumen membrane. Using a reporter for PIP2, PH:GFP, we saw more diffuse signal in the lateral region in animals depleted of WAVE component *gex-3*. In addition, while PH::GFP is normally apically enriched, this apical enrichment and total membrane accumulation dropped in animals depleted of the WAVE complex (Figure 3). Since these animals still showed adhesion at apicolateral and basolateral regions, the adhesive barrier appeared intact. However, these changes suggested a reduction in membrane polarity that normally enriches PIP2 apically. Loss WAVE or Cadherin reduced the PIP2 biosensor levels, especially at apical regions. Since PIP2 is known to reinforce apical/basal polarity [28,29], this result suggested an inter-dependent role for Cadherin, WAVE complex, and PIP2 in reinforcing apical/basal polarity. One mechanism for this integration may be through PIP2′s role promoting polarized transport of proteins [28]. 

Previous studies suggested a second role for branched actin at apical junctions: to continually push lateral membranes together in adult tissues to repair small breaks. In our system, we did not see evidence that animals under laboratory conditions experience lateral breaks in their intestinal cells. However, *C. elegans* intestinal cells are not replaced if they are injured. Therefore, animals may need to repair altered or damaged membranes when they are challenged by pathogens, or when they ingest harmful chemical products. Our studies suggest that there is a dynamic and polarized pool of Cadherin present throughout the life of the worm, and that branched actin is active in stabilizing and polarizing this dynamic pool. This suggests that even in cells that are not migrating, or obviously damaged, in an animal that is constantly moving, and therefore putting these cells under some strain, branched actin supports and stabilizes a pool of dynamic Cadherin at the apicolateral junction. In the future, it will be interesting to investigate if different environmental conditions affect the requirements for Cadherin and branched actin in the mature intestine, and to explore mechanisms that promote branched-actin dependent polarized enrichment of Cadherin/HMR-1. 

## Figures and Tables

**Figure 1 jdb-09-00019-f001:**
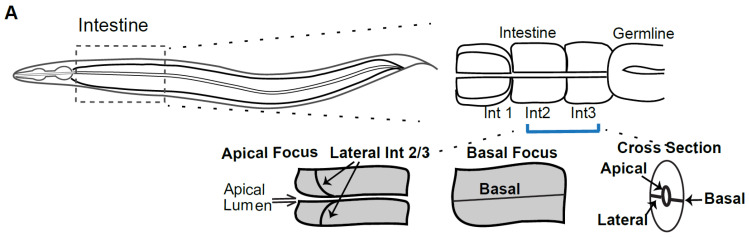
Cadherin localization is polarized and this depends on WAVE. (**A**) *Upper panel:* Cartoon of a *C. elegans* worm, oriented anterior to posterior, with an emphasis on the intestine, which begins right behind the pharynx (bilobed structure at the anterior) and extends for the length of the worm. The region within the dotted line is where most of the imaging and measurements in this study were conducted, especially at the boundary between intestinal rings 2 and 3 (Int 2/3). Ring 1 has four cells, while rings 2–8 have two cells each. *Lower panel*: Cartoons to illustrate how a typical region of the intestine, with two cells, is oriented. Since worms are usually imaged on their side, rings 2 and 3 have a basal regions at the top surface (Basal Focus), while focusing at the center of the intestine reveals the apical region, including the lumen, and the lateral regions where two rings meet. (**B**) Cadherin/HMR-1 endogenously tagged with GFP using CRISPR [23] is expressed post-embryonically in various epithelia. Parallel yellow arrows indicate the apical lumen, which shows low enrichment of HMR-1::GFP. Yellow solid arrow indicates the lateral junction between rings 2 and 3. *Center panel:* The levels of HMR-1::GFP were measured at apicolateral, midlateral and basolateral regions. The branched actin regulator GEX-3 was depleted with two day RNAi feeding treatment, and the same regions were compared. Right 2 panels: Crop of representative ring 2/3 lateral regions, including a Z-projection from 4 slices, 0.4 μm apart, to better illustrate the differences between controls and animals depleted of *gex-3.* All images were obtained on a Zeiss AxioImager Z1 with a Yokogawa CSUX1-5000 spinning disc with a Plan Apo 63X/1.4NA Oil lens. All worms shown in this and other figures are oriented with anterior to the left. Scale bars shown in this and all figures are 5 μm long unless otherwise stated. (**C**) Mean and maximum intensities of HMR-1::GFP at each region of the lateral membrane. For this and all figures, the Line Scan tool of ImageJ was used to compare fluorescent intensities. For these measurements, a line of 35 pixels (3 μm) was drawn in the lateral region indicated and both mean and maximum intensities were recorded. Each animal was measured at the Int2/3 lateral membrane, some on one side, and some on both sides if both were in focus. Seven animals were used for the controls, and eight for *gex-3* RNAi. Statistical analysis was performed with Graphpad Prism. For this and other figures, error bars show standard error of the mean (SEM). Asterisks mark statistical significance, * = *p* < 0.05, ns = not significant.

**Figure 2 jdb-09-00019-f002:**
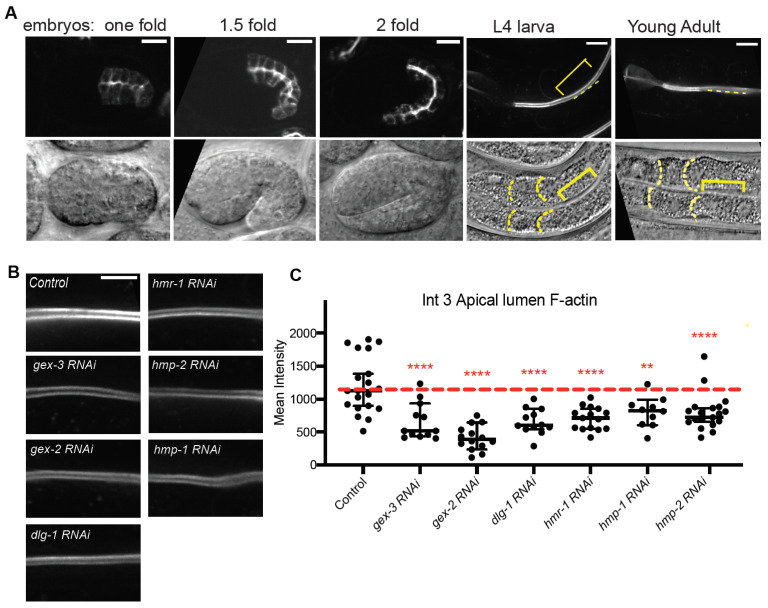
Cadherin complex and WAVE complex support apical F-actin. (**A**) Developmental expression of the new transgene, *Pglo-1::Lifeact::T**AG::RFP*, built for this study. Expression begins in embryos and is enriched apically by the one-fold embryonic stage. It becomes increasing enriched apically, as shown in L4, the late larval stage used for most of this study, to young adults, one day older than L4. Brackets indicate the apical region of ring Int3, and the dotted lines indicate how apical lumen enrichment was measured in B. All scale bars shown are 5 μm long, except for panel (**E**). (**B**) Representative images of apical F-actin in intestinal ring 3 in L4 larvae in controls and in animals depleted by feeding RNAi of regulators of branched actin, or components of the apical junction. (**C**) Mean intensity of F-actin in Int3. A line scan of 80 pixels (7 μm) was used to measure the average F-actin enrichment at the apical lumen of ring Int3. Each data point represents one animal: 11 for control, 7 for *gex-3 RNAi*, 7 for *gex-2 RNAi*, 7 for *hmr-1 RNAi*, 9 for *dlg-1 RNAi*, 5 for *hmp-1 RNAi*, 12 for *hmp-2 RNAi*. (**D**) While most of the F-actin is found along the apical lumen (shown at bottom in (**D**) and (**E**)), there is enrichment along the lateral membrane (white arrows), compared in controls, and in animals depleted of WAVE complex component *gex-2* and Cadherin components. The frequency of lateral abnormalities (yellow asterisks) and F-actin changes is listed below each image. “bubbles” refers to vesicle-like enrichments at the lateral membrane in *hmp-1* RNAi. “Reduced” refers to the very low lateral signal in *gex-2* RNAi. The filled-in vesicles seen in the *gex-2 RNAi* image are likely lysosomes, which tend to take up mCherry fluorescence. The actual lateral membrane is so reduced it is almost invisible. (**E**) In embryonic intestines, WAVE components are enriched more apically than the Cadherin Junction components [1]. Here, two CRISPR strains were combined to compare endogenous enrichment of HMR-1::mKate2 and GFP::FLAG::GEX-3. Yellow arrow heads point to the apicolateral membrane. Scale bar in this panel is 2.5 μm. ** = *p* < 0.005, **** = *p* < 0.0005, ns = not significant.

**Figure 3 jdb-09-00019-f003:**
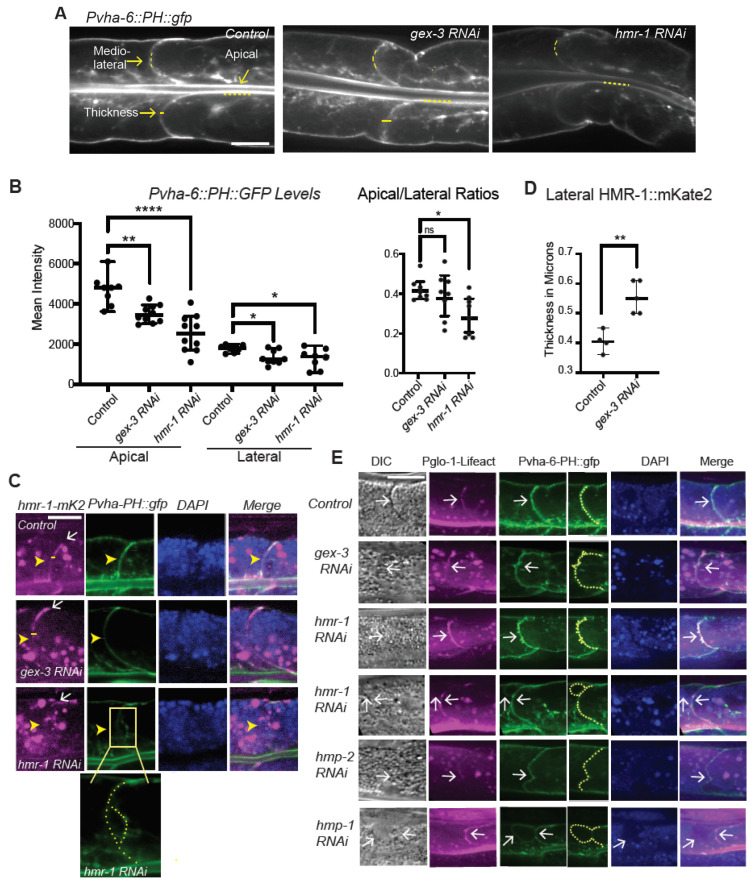
Cadherin lateral morphology depends on WAVE. (**A**) *Pvha-6::PH::GFP* labels the intestinal membrane, is a biosensor for PIP2, and can show the morphology of the cells. L4 controls and animals depleted of *gex-3* or *hmr-1* with three days RNAi feeding were compared for enrichment at apical and lateral regions. Scale bars are 5 μm long for all panels. (**B**). Mean intensity of the apical (luminal) and midlateral region of the intestinal ring 3 membrane. Each data point represents one or two measurements per animal: 5 Control, 5 *gex-3 RNAi* and 5 *hmr-1 RNAi* animals. Line scans of 80 pixels (7 μm) and 35 pixels (3 μm) were used to measure the lumen and midlateral membrane signal, respectively, indicated by the dotted yellow lines in (**A**). (**C**) The lateral membrane of Int3 was compared by monitoring Cadherin (*hmr-1::mKate2*) and PIP2 *(Pvha-6::PH::gfp*) on one side of the intestine, with all images in (**C**) and (**D**) oriented so that apical is down and basal is up. White arrows point to midlateral regions. (**D**) The thickness of the midlateral membrane region enriched in *hmr-1::mKate2* was measured, as indicated by the yellow bars in panel A and C. Each data point represents one animal: 4 Controls, 5 *gex-3 RNAi* animals. (**E**) The lateral membrane of Int3 was further compared by monitoring a strain with both intestinal F-actin (*Pglo-1::Lifeact::TAG-RFP*) and intestinal PIP2 (*Pvha-6::PH::gfp).* White arrows point to the lateral membrane and regions of altered F-actin or PIP2, including membrane distortions. * = *p* < 0.05, ** = *p* < 0.005, **** = *p* < 0.0005, ns = not significant.

**Figure 4 jdb-09-00019-f004:**
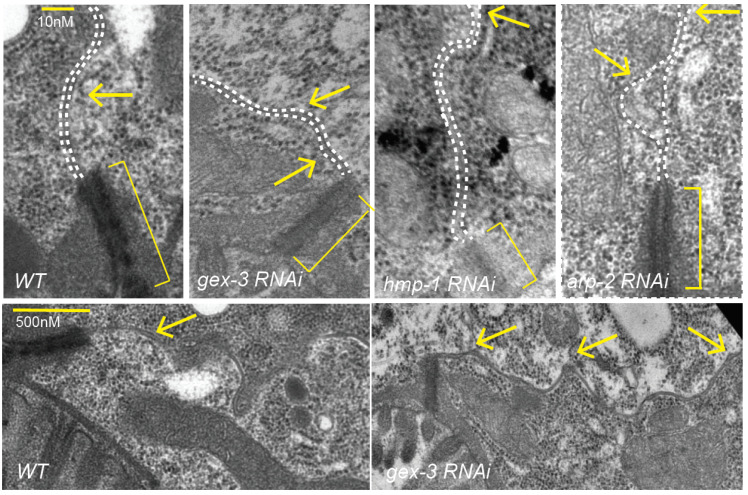
Electron micrographs (EM) analysis of adult intestines show effect of reducing *gex-3* or *arp-2* on lateral membranes. Electron micrographs to compare apical and lateral regions of the adult intestine in control animals and animals depleted of components of the WAVE complex (*gex-3*), Cadherin complex (*hmp-1*) or Arp2/3 complex (*arp-2*) via RNAi feeding. Brackets indicate the electron-dense apical junction. Arrows point to the lateral membrane between two intestinal cells, including regions of membrane separation. Dotted white lines follow the lateral membranes between two intestinal cells. * = *p* < 0.05, ** = *p* < 0.005, *** = *p* < 0.0005, ns = not significant.

## Data Availability

Data is contained within the article or Appendix A.

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
