# Peer review of "E-Cadherin/HMR-1 Membrane Enrichment Is Polarized by WAVE-Dependent Branched Actin"

_jdb, 2021, doi:10.3390/jdb9020019_

Round 1
Reviewer 1 Report
The authors use a developmental animal model (C. elegans embryo) to study the apical junction enrichment of E-cadherin receptors and their connection to apical F-actin as regulated to by WAVE-dependent modulation of branched actin. They suggest relevance for cancer cell polarity.
The analysis is based on state-of-the-art techniques using RNAi silencing and life imaging of C elegans embryos based on: (1) GFP-labelled E-cadherin, using the CRISPR-engineered endogenous gene and (2) fluorescently labelled sensors for F-actin (lifeact) and PIP2 (PH domain of rat PLC-delta)
Comments and suggestions:
(1) The reviewer sees the following conceptual problems:
(i) Apical junctions in epithelia are defined by tight junction (TJ) receptors (or zonula occludens) leading to apicobasal compartmentalization of epithelia and to polarity. Adherens junctions (AJs) formed by cadherin receptors are localized underneath or lateral to TJs, not apical. If this is different in C. elegans epithelia (figure 4 suggests otherwise), this should be addressed. Otherwise, consideration of TJ involvement should be crucial to the interpretation of data presented in this study. (In case C. elegans should be fundamentally different, a comparison to mammalian epithelia is difficult or even irrelevant.)
(ii) E-cadherins are homotypic transmembrane adhesion receptors. There are no binding partners (for these receptors) in the apical (lumen) nor basal (ECM) membrane (with no adjacent epithelial cells) and therefore, no functional adhesion structures allowing force transduction to and from microfilaments.
(iii) E-cadherin can be physically connected to contractile filaments via adhesion complexes involving catenins and further adaptor proteins. The connection to / regulation by branched actin filaments is not understood at the level of protein-protein interactions.
These points should be more clearly covered / discussed in introduction and/or discussion
(2) Interpretation of lateral defects and their relevance are not adequately discussed:
Please define the terms (i) width of membrane and (ii) polarity of membrane
Fig 3A/D does “width of membrane “ mean separation of membranes due to destabilization of adherens junctions as suggested by Fig 4 ?
Fig 3A/C/E “Change of PIP2 intensity”, does this signify leakiness of the apical junction or alteration in PIP2 production / stability? If the junction became leaky, reduction of apical signal should lead to increased lateral signal, which is not observed. Lines 244/245
What is the proposed mechanism behind the change in PIP2-detection?
Fig 2D. [3/8 “bubbles”] AND [9/9 reduced]
Should images (fluorescent signal) or legends be swapped?
Do these observations in Fig 2D indicate instability of the lateral cell-cell junction?
(3) Measurements for quantitative analyses (fig. 1C, fig. 2C, fig 3 B/D):
Please clearly state number of animals analyzed (and images per animal)
So far, the only hint is given in lines 363/364: “Each dot represents one measurement, 7 Ctrl and 8 gex-3animals”. However, dots per condition in fig. 1C vary between 10-13.
Were several images of some animals used?
No details are given for usage of animals in quantification of fig. 2C and fig 3 B/D
(4) methods:
Description and usage of C. elegans strains is detailed at best rudimentary:
e.g. fig 2E (three dis-agreeing descriptions!)
label: Pvha-6-PH::gfp; line 383: details GFP::WVE-1 and line 336 suggests OX974 strain
Controls for “of target effects” of RNAi experiments are not discussed; no information on RNAi sequences is given
(5) wording:
CRISPR tagged E-Cadherin (line 288) and similar.
Description may be derived from context of the manuscript, but there are no CRISPR-tagged proteins in your analysis.
Author Response
Reviewer 1:
The authors use a developmental animal model (C. elegans embryo) to study the apical junction enrichment of E-cadherin receptors and their connection to apical F-actin as regulated to by WAVE-dependent modulation of branched actin. They suggest relevance for cancer cell polarity.
The analysis is based on state-of-the-art techniques using RNAi silencing and life imaging of C elegans embryos based on: (1) GFP-labelled E-cadherin, using the CRISPR-engineered endogenous gene and (2) fluorescently labelled sensors for F-actin (lifeact) and PIP2 (PH domain of rat PLC-delta)
Comments and suggestions:
(1) The reviewer sees the following conceptual problems:
(i) Apical junctions in epithelia are defined by tight junction (TJ) receptors (or zonula occludens) leading to apicobasal compartmentalization of epithelia and to polarity. Adherens junctions (AJs) formed by cadherin receptors are localized underneath or lateral to TJs, not apical. If this is different in C. elegans epithelia (figure 4 suggests otherwise), this should be addressed. Otherwise, consideration of TJ involvement should be crucial to the interpretation of data presented in this study. (In case C. elegans should be fundamentally different, a comparison to mammalian epithelia is difficult or even irrelevant.)
(ii) E-cadherins are homotypic transmembrane adhesion receptors. There are no binding partners (for these receptors) in the apical (lumen) nor basal (ECM) membrane (with no adjacent epithelial cells) and therefore, no functional adhesion structures allowing force transduction to and from microfilaments.
(iii) E-cadherin can be physically connected to contractile filaments via adhesion complexes involving catenins and further adaptor proteins. The connection to / regulation by branched actin filaments is not understood at the level of protein-protein interactions.
These points should be more clearly covered / discussed in introduction and/or discussion
Response: The Reviewer raises important points that we address in various sections of the revised manuscript:
We have revised the introduction to better introduce similarities and differences between C. elegans apical junctions, and vertebrate ones. For example, C. elegans has a single electron dense junctional complex, compared to two in vertebrates and flies (Reviewed in Pasti and Labouesse, 2016). Also, the tight junction component ZOO-1, the homolog of vertebrate ZO-1, is recruited to the Cadherin junction by HMR-1/E-Cadherin, but it does not appear to be required for paracellular gate function (Simske et al., 2003; Lockwood et al., 2008). Finally, as is well explained in several excellent reviews, the apical-most complex in C. elegans is the Cadherin complex, not a separate Tight Junction. This was added to the Introduction, and explains why we focus on Cadherin, and not tight junction proteins.
- elegans apical junctions are different, though the data suggest the Cadherin complex molecules play similar roles, at a functional, biochemical and structural level (See for example, Kwiatkowski et al.., 2010 DOI: 10.1073/pnas.1007349107 and Shao et al., 2017 DOI: 10.1074/jbc.M117.795567 ). To avoid confusion we revised the last sentence in the Abstract to say “cell polarity” rather than “cancer cell polarity,” so we don’t exaggerate the implication for mammalian studies.
(2) Interpretation of lateral defects and their relevance are not adequately discussed:
Please define the terms (i) width of membrane and (ii) polarity of membrane
Response: We have redefined the “width” as the Lateral membrane “thickness”. We added these measurements since a significant number of the mutant lateral membranes appeared both more diffuse and wider. We noticed the difference was statistically significant.
“polarity” of the membrane is redefined as…. “Apical/basal polarity”. The segregation of lipid domains marked by PIP2 and other lipid markers like PIP3 is required for homeostasis, and requires that the cell maintain membrane subdomains. The introduction now introduces phosphoinositides and their role supporting apical/basal polarity, and the interdependence of lipid polarity and apical/basal polarity.
Fig 3A/D does “width of membrane “ mean separation of membranes due to destabilization of adherens junctions as suggested by Fig 4 ?
Response: Yes, we believe this is the case. The images in Figure 3 may have been too small to appreciate the separations. Therefore we have added a few zoomed in views of the same images, and added dotted lines to delineate exactly where we see membrane separations.
Fig 3A/C/E “Change of PIP2 intensity”, does this signify leakiness of the apical junction or alteration in PIP2 production / stability? If the junction became leaky, reduction of apical signal should lead to increased lateral signal, which is not observed. Lines 244/245
Response: The Reviewer is correct that we are seeing both reduction in overall PIP2, and a change in the apical enrichment, but the more lateral regions do not increase, instead showing similar or lower levels. The largest effect is at the apical regions. To better measure this, we have added the ratio of apical to lateral for controls and mutants. These are added to Fig. 3 and show there is also a significant change in the ratio for animals depleted of Cadherin/hmr-1, and less so for animals depleted of the WAVE component, gex-3.
What is the proposed mechanism behind the change in PIP2-detection?
Response: Loss of apical/basal polarity would reduce lipid polarity. We have added to the discussion that one possible mechanism would be through loss of apicolateral Cadherin. If Cadherin supports the membrane polarity, loss of properly enriched apicolateral Cadherin might interfere with apical retention of PIP2. If WAVE and Cadherin are maintaining apical/basal polarity by maintaining the apical junction, loss of these proteins would be expected to reduce the segregation of the lipid domains. Once phosphoinositides fail to be recruited to the correct compartment, they may be lost from the membrane. A more mechanistic explanation is beyond the scope of this study. Several studies point to a possible role for PIP2 promoting actin regulation at Focal Adhesions (see for example, Leaget et al., EMBO J 2011, 30:4539-4553. http://dx.doi.org/10.1038/emboj. In addition, polarity regulators like Crumbs have been shown to regulate the levels of apical PIP2 (Lattner et al., eLife: 2019 Nov 7;8:e50900.
doi: 10.7554/eLife.50900).
Fig 2D. [3/8 “bubbles”] AND [9/9 reduced]
Should images (fluorescent signal) or legends be swapped?
Response: No, but thank you for asking since it suggests the changes are not obvious and need better labeling, and description. We have added clarifications to the figure legends and text to explain that:
The “bubbles” refers to vesicle-like enrichments at the lateral membrane in hmp-1 RNAi. “Reduced” refers to the very low lateral signal in gex-2 RNAi. The filled-in circles seen in the gex-2 RNAi image are likely lysosomes, which appear to take up the mCherry fluorescence when F-actin is lost from the lateral membrane. The actual lateral membrane is so reduced it is almost invisible. We hope the extra images with dotted lines make these changes easier to see.
Do these observations in Fig 2D indicate instability of the lateral cell-cell junction?
Response: Yes, that is one interpretation of these lateral membrane changes.
(3) Measurements for quantitative analyses (fig. 1C, fig. 2C, fig 3 B/D):
Please clearly state number of animals analyzed (and images per animal)
So far, the only hint is given in lines 363/364: “Each dot represents one measurement, 7 Ctrl and 8 gex-3animals”. However, dots per condition in fig. 1C vary between 10-13.
Were several images of some animals used?
No details are given for usage of animals in quantification of fig. 2C and fig 3 B/D
Response: We have added the exact number of animals used for each measurement to the Figure legends for Figures 1-3. We have added the exact number of animals measured, and how many measurements per animal, one or two: sometimes we measure both Int 2/3 lateral membranes, both top and bottom lateral junction, since there are two cells per intestinal ring, with two lateral membranes in Int 2/3.
(4) methods:
Description and usage of C. elegans strains is detailed at best rudimentary:
e.g. fig 2E (three dis-agreeing descriptions!)
label: Pvha-6-PH::gfp; line 383: details GFP::WVE-1 and line 336 suggests OX974 strain
Response: Thank you for catching this. The Figure Legend was correct. The labeling in figure 2E has been fixed. The point of showing this was to ask where GFP::GEX-3 and HMR-1::mKate2 are most enriched at the lateral membrane. The highest expression is clearly at the apicolateral region.
Controls for “of target effects” of RNAi experiments are not discussed; no information on RNAi sequences is given
Response: We have expanded the Methods to explain how RNAi was done and monitored.
In addition, we added a Supplemental Figure including a table to document % embryonic lethality, our usual strictest test for RNAi strength. Since the changes we are monitoring are in adult intestines we also imaged silencing of a transgene in the intestine, for each gene being depleted by RNAi. This is shown in a small panel added to the Supplemental Figure. We will leave it up to the Editor and Reviewers to decide if they want to include this Supplemental Figure. The % lethality and % drop in GFP signal is also mentioned in the Methods.
(5) wording:
CRISPR tagged E-Cadherin (line 288) and similar.
Description may be derived from context of the manuscript, but there are no CRISPR-tagged proteins in your analysis.
Response: We think we understand the confusion. All of the apical junction proteins in this manuscript are CRISPR-tagged, meaning the endogenous gene was tagged in its naturally occurring chromosomal position using CRISPR technology (Marston et al., 2016). This sometimes give fainter expression than over-expression lines previously used, and is preferable since it more accurately reflects the expression of the protein in a live animal. Since the term “CRISPR-tagged” may confuse readers, we revised the description of the strains when they are first introduced, and try to be more consistent in how we label them. Other strains built for this study, like the intestinal F-actin strain, Pglo-1::Lifeact::mCherry, is a more traditional “transgene” where we injected DNA, then integrated it at random in a chromosome, where we expect it formed a multi-copy array.
Reviewer 2 Report
This manuscript provides novel characterization of E-cadherin in the adult C. elegans intestine. In particular, they use a endogenously-tagged probe to demonstrate the apicolateral localization of E-cadherin in the anterior gut, and show that this localization depends on polarity and branched F-actin. This could reveal a novel mechanism whereby branched F-actin influences the turnover of E-cadherin in polarized cells.
While the manuscript offers some novel insight to the regulation of E-cadherin, there are several aspects that should be addressed.
Major comments:
- The introduction is somewhat disorganized. It is not clear what has been done before vs. what is shown in this manuscript. For example, Lines 65-69 and 75-79. What is the goal of the study? This needs to be more clear to the reader. Clarify the mode that is being tested.
-
Figure 1B – It is not clear 1) how the authors know where the apical vs. lateral (e.g. between 2-3 cell) and basal boundaries are, and 2) that these are intestinal cells. They should consider using markers to co-stain for both, and then overlay the E-cadherin signal. If the cells have shifted in z in treatments vs. control, then it is not clear what is really being measured.
-
One major concern is the extent of knockdown by RNAi. It is not mentioned in the text how the RNAi was done until the later figures, where it indicates 2 (or 3) days. This does not provide the reader with meaningful information. They need to provide more information as to how this was done. But more importantly, they should measure the extent of knockdown – what were the levels of the different proteins like at the time they are analyzing phenotypes? Also, they should address how they are able to overcome requirements during embryogenesis, and if it is possible that these earlier requirements cause indirect changes in the adult intestine. The interpretation of data should consider both partial knockdown (vs. complete) and that earlier phenotypes could contribute to those seen in the adult. Are there zygotic alleles that they could use that may permit them to analyze phenotypes more specifically in the adult?
- For quantitation, over what area were pixels measured, and how were levels normalized to account for bleaching, or other factors that cause variability? Especially when comparing RNAi conditions to controls, ratios may be more informative vs. mean intensity. For example, for Figure 2C – this could be quantified as a relative ratio at the apical vs. other boundaries. Especially if there is a tendency for levels to increase or become less uniform laterally/basally, and this would better control for signal variabilities from worm-to-worm. This was done for the biosensor (Lines 242-3), but should be done for all figures and especially experimental conditions describing apico/lateral enrichments. This is essential to support statements such as ‘The largest change was at apical membranes..’.
Minor edits/comments:
Introduction
C. elegans is not in italics.
Line 39 should be "branched actin" instead of "branch actin".
Line 55 “this complex results” should be "these results" or "this result"
Line 89-90 – larger than what?
Introduce some of the polarity markers/membrane lipids in the introduction, so it won’t seem out of place in the results. Also, there will be a logical flow from testing the loss of apical enrichment of actin to change in polarity.
Materials and Methods
Need to include more information for light microscopy and EM. Readers may not have access to prior published papers and methods should be stand-alone. Also, some methods in the figure legends, which should be moved to this section.
Indicate where strains come from.
Formatting issues.
Results
Developmentally, how do int 2 and 3 cells compare vs. the posterior gut cells? Differential promoter requirements suggests differences in cell types. Was E-cadherin also observed in these cells? It doesn’t need to be shown per se (I appreciate challenges in imaging when the intestine goes behind the germline), but would be good to include a description.
Lines 174-179 – the text is confusing. It reads as if the signal is apicolateral, then says apically enriched, but then says not at lumen (which should be apical) and is at basal membranes. Which is it? Or is something missing regarding cell type?
Line 204 – the word actin is repeated.
Figure 2E – referred to as gfp::gex-3 (main text), GFP::WVE-1 (legend) and Pvha-6-PH::gfp – which one is it? The authors indicate that they intended to show co-localization with WAVE and HMR-1.
Lines 242-3 – this is the first time it is mentioned how RNAi was done. First, this information should be in the methods. But more importantly, this is not very helpful to readers. They need to know more information about how the knockdown was done, and the extent of knockdown (see comment above). My interpretation is that weak phenotypes are a reflection of partial vs. complete knockdown.
The description of the double (e.g. hmr-1::mKate2 and the biosensor) is confusing – at some points referring to just hmr-1, but at others referring to both in a more relative way.
It is not clear to the reader what the increase in lateral junction width reflects. I understand that the authors do not want to speculate in the results, but the rationale/logic for why this was measured is missing. The changes in lateral junctions are reminiscent of those previously described for the lateral junctions of epidermal cells in the embryo by the Zaidel-Bar (e.g. SRGP-1) and Labouesse labs (e.g. RGA-2). I don’t think width is the correct measurement to use for this. These could reflect differences in ‘pulling’ events in the gex-3 RNAi, which makes sense – if long, unbranched filaments dominate, there could be more force/tension generated at the junction. This is different from the hmr-1 RNAi where F-actin is less uniform, likely because of the decreased organization of F-actin at the junction.
Figure 3A) consider showing the images with and without the pseudocoloring/yellow lines on top (especially width of gex-3 RNAi).
Figure Legend for Figure 3 - Line 391 says yellow arrows but they’re white.
Figure 4 – the membrane in hmr-1 RNAi is not visible, and some of the out-pocketings/separations in the lower gex-3 RNAi panel do not appear different vs. control. Also, consider measuring the change in electron-dense junctions to support the statement in the text. It is hard to know what is meant by lateral expansion, especially when only showing a small portion of the cell membrane. The mild phenotypes are likely due to mild RNAi.
Discussion
Formatting issues.
Lines 284-286- says puzzle twice, edit to remove one.
Combine Lines 284-300 into one paragraph.
Line 293 – confusing as written. The text describes the apical localization of cadherin, but then basal localization is emphasized. In the figure, arrows point to apical enrichments. I recommend describing what was seen first (summary of results), then tell us how this 1) fits with prior results and models, and 2) doesn't. For example, apicolateral localization may be expected based on E-cadherin localization in other cell types, but its basal localization may be unexpected.
Line 307 – when referring to the biosensor, states ‘widening of lateral regions’ – but this wasn’t measured with this probe.
Switching between gex-3 and WAVE complex – be consistent.
Line 310 “since these animals stills ….” Should be still.
Line 321-322 “branched actin is supports …” remove is.
Clarify the model. Is it that branched F-actin could control trafficking, which could maintain a balance of cadherin at the apicolateral region? Or is it by competing with linear filaments to balance their number? Or are they providing different forces at junctions? It isn’t clear what the role of branched F-actin is proposed to be, and it would help the reader if the authors could speculate.
Author Response
This manuscript provides novel characterization of E-cadherin in the adult C. elegans intestine. In particular, they use a endogenously-tagged probe to demonstrate the apicolateral localization of E-cadherin in the anterior gut, and show that this localization depends on polarity and branched F-actin. This could reveal a novel mechanism whereby branched F-actin influences the turnover of E-cadherin in polarized cells.
While the manuscript offers some novel insight to the regulation of E-cadherin, there are several aspects that should be addressed.
Major comments:
- The introduction is somewhat disorganized. It is not clear what has been done before vs. what is shown in this manuscript. For example, Lines 65-69 and 75-79. What is the goal of the study? This needs to be more clear to the reader. Clarify the mode that is being tested.
Response: We have revised and streamlined the introduction.
What is new:
Where exactly does C. elegans Cadherin localize in a postembryonic tissue; Where is it relative to WAVE? What happens in WAVE KD? Is F-actin collectively localized by WAVE and Cadherin? Is the polarity of membranes supported by Cadherin and WAVE? What happens to lateral membrane when there is Cadherin or WAVE KD? Suggested mechanism: not addressed here, but the appearance of membrane vesicles by live imaging and EM suggests altered transport of proteins may explain the dependence of Cadherin on branched actin for apical enrichment and maintenance.
- Figure 1B – It is not clear 1) how the authors know where the apical vs. lateral (e.g. between 2-3 cell) and basal boundaries are, and 2) that these are intestinal cells. They should consider using markers to co-stain for both, and then overlay the E-cadherin signal. If the cells have shifted in z in treatments vs. control, then it is not clear what is really being measured.
Response: Some of the markers we show here, like Pglo-1::Lifeact::mCherry, are only expressed in the intestine. For anyone trained properly in viewing C. elegans tissues it is simply impossible to mistake the intestine with other tissues. The intestinal cells look quite different from the basal view, in contrast to the central-focus, apical view shown for most of this paper. We included DIC (shown in Fig. 2 &3), and have been imaging these cells for quite some time (Bernadskaya et al., 2011) so there is no possibility of confusing apical and basal views in the intestine. For the purpose of making it easier on readers, we previously only showed the apical view of the intestine, but now we have added basal views of these cells to Figure 1B to help orient readers.
- One major concern is the extent of knockdown by RNAi. It is not mentioned in the text how the RNAi was done until the later figures, where it indicates 2 (or 3) days. This does not provide the reader with meaningful information. They need to provide more information as to how this was done. But more importantly, they should measure the extent of knockdown – what were the levels of the different proteins like at the time they are analyzing phenotypes? Also, they should address how they are able to overcome requirements during embryogenesis, and if it is possible that these earlier requirements cause indirect changes in the adult intestine. The interpretation of data should consider both partial knockdown (vs. complete) and that earlier phenotypes could contribute to those seen in the adult. Are there zygotic alleles that they could use that may permit them to analyze phenotypes more specifically in the adult?
Response: We have expanded the explanation for how RNAi was done, and added a Supplemental figure. Briefly, for analyzing adult phenotypes, we feed L1 larvae for 2-3 days on bacteria expressing specific dsRNA for the gene being studied. So the L4 larvae and young adults imaged are animals past embryogenesis, but exposed during post-embryonic growth. For some genes, like arp-2 RNAi, 2 days is the longest we can go before the animals become extremely sick. For most genes, since RNAi does not remove existing proteins (due to protein stability), it reduces proteins, but not 100%. Some genes are easy to knock down (gex-3) while others, even ones that are part of the same complex (like wve-1) are resistant to RNAi for reasons that have yet to be established. In the future we hope to adapt degron technology, which would allow rapid and transient depletion, a clear next step in gene expression studies.
- For quantitation, over what area were pixels measured, and how were levels normalized to account for bleaching, or other factors that cause variability? Especially when comparing RNAi conditions to controls, ratios may be more informative vs. mean intensity. For example, for Figure 2C – this could be quantified as a relative ratio at the apical vs. other boundaries. Especially if there is a tendency for levels to increase or become less uniform laterally/basally, and this would better control for signal variabilities from worm-to-worm. This was done for the biosensor (Lines 242-3), but should be done for all figures and especially experimental conditions describing apico/lateral enrichments. This is essential to support statements such as ‘The largest change was at apical membranes..’.
Response: In Fig. 2C we decided to report apical measurements of the 3rd intestinal ring, Int 3, which gave consistent levels in Control, significantly higher than in the mutants. We always see higher signal in Int2 apical lumen, relative to Int 3. Comparing controls and mutants in Int2 also showed a drop in the mutants. We don’t see a shift to lateral regions. Instead the change is a drop in the apical lumen. We tried to also measure changes in lateral Pglo but it was so low in the mutants we did not think it would change our findings.
Minor edits/comments:
Introduction
- elegans is not in italics.
Response: Thank you this was fixed.
Line 39 should be "branched actin" instead of "branch actin". Response: Thank you, fixed.
Line 55 “this complex results” should be "these results" or "this result"
Response: It now says “these complex results”
Line 89-90 – larger than what?
Response: We added:,” relative to embryonic cells,”
This was the motivation for switching from embryos to adults: bigger cells and some new imaging tools.
Introduce some of the polarity markers/membrane lipids in the introduction, so it won’t seem out of place in the results. Also, there will be a logical flow from testing the loss of apical enrichment of actin to change in polarity.
Response: Thank you for this suggestion. We have reorganized the introduction, and these terms and polarity markers are introduced earlier.
Materials and Methods
Need to include more information for light microscopy and EM. Readers may not have access to prior published papers and methods should be stand-alone. Also, some methods in the figure legends, which should be moved to this section.
Response: We have expanded the Methods to include more microscopy and EM details, and shifted some details from Figure Legends to Methods. However, we prefer having informative Figure Legends.
Indicate where strains come from.
Response: We have added References to the Methods to address this.
Formatting issues.
Results
Developmentally, how do int 2 and 3 cells compare vs. the posterior gut cells? Differential promoter requirements suggests differences in cell types. Was E-cadherin also observed in these cells? It doesn’t need to be shown per se (I appreciate challenges in imaging when the intestine goes behind the germline), but would be good to include a description.
Response: Yes, all of the intestinal cells express Cadherin. The only reason we showed just these two cells was to simplify the story, and to always compare the same region of the intestine. Even when a protein is expressed all throughout the intestine, it is clear some regions have higher or lower expression.
Lines 174-179 – the text is confusing. It reads as if the signal is apicolateral, then says apically enriched, but then says not at lumen (which should be apical) and is at basal membranes. Which is it? Or is something missing regarding cell type?
Response: We agree this was confusing. We have reworded this paragraph to say,
“endogenously tagged Cadherin/HMR-1 is apically enriched within the lateral membrane (apicolaterally) in larval and adult intestines. By comparison, there is low Cadherin enrichment along the lumen, while a surprising amount of Cadherin is enriched at basal membranes (Figure 1 B).”
The basal amounts are lower than at apical regions, but we were surprised how much endogenously tagged Cadherin is found at the basal membrane. In models for how Cadherin is delivered to the apicolateral membrane, some propose it is first trafficked to basal membranes, and then moves apically (Woichansky et al., 2016), so this basal population may reflect a pool delivered basally. We hope to make movies in the future to watch how Cadherin moves over time, to test if there is both apicolateral and basolateral transport, as predicted by those models. Live imaging of this transport has not been documented, as far as we know.
Line 204 – the word actin is repeated.
Response: Thank you, the second “F-actin” was replaced with the name of the new F-actin strain, Pglo-1::Lifeact::TAG-RFP.
Figure 2E – referred to as gfp::gex-3 (main text), GFP::WVE-1 (legend) and Pvha-6-PH::gfp – which one is it? The authors indicate that they intended to show co-localization with WAVE and HMR-1.
Response: Thank you for catching this. The text was correct, and now the Figure legend and Figure labeling are correct: this is strain OX974 which combines hmr-1::mKate2 with gfp-FLAG-gex-3, both CRISPR labeled alleles from the Goldstein lab (Marston et al., 2016).
Lines 242-3 – this is the first time it is mentioned how RNAi was done. First, this information should be in the methods. But more importantly, this is not very helpful to readers. They need to know more information about how the knockdown was done, and the extent of knockdown (see comment above). My interpretation is that weak phenotypes are a reflection of partial vs. complete knockdown.
Response: We expanded the Methods on RNAi, added to the text how we monitored knock-down, and we added a Supplemental Figure to address this. So yes, we are seeing interesting lateral membrane changes even when RNAi knockdown is partial.
The description of the double (e.g. hmr-1::mKate2 and the biosensor) is confusing – at some points referring to just hmr-1, but at others referring to both in a more relative way.
Response: We revised the writing to explain…
It is not clear to the reader what the increase in lateral junction width reflects. I understand that the authors do not want to speculate in the results, but the rationale/logic for why this was measured is missing. The changes in lateral junctions are reminiscent of those previously described for the lateral junctions of epidermal cells in the embryo by the Zaidel-Bar (e.g. SRGP-1) and Labouesse labs (e.g. RGA-2). I don’t think width is the correct measurement to use for this. These could reflect differences in ‘pulling’ events in the gex-3 RNAi, which makes sense – if long, unbranched filaments dominate, there could be more force/tension generated at the junction. This is different from the hmr-1 RNAi where F-actin is less uniform, likely because of the decreased organization of F-actin at the junction.
Response: We expanded the introduction to explain that one predicted role for branched actin at the membrane is to push the lateral junctions together, even in a mature epithelium. The apparent widening of the lateral membrane, which by EM reveals membrane separations, supports this role. In this model, branched actin would be creating force to keep the lateral junction closely adherent.
Figure 3A) consider showing the images with and without the pseudocoloring/yellow lines on top (especially width of gex-3 RNAi).
Response: We removed one of the yellow lines, so the other side of the lumen is easier to see.
Figure Legend for Figure 3 - Line 391 says yellow arrows but they’re white.
Response: Thank you, this was fixed in the figure legend of Figure 3.
Figure 4 – the membrane in hmr-1 RNAi is not visible, and some of the out-pocketings/separations in the lower gex-3 RNAi panel do not appear different vs. control. Also, consider measuring the change in electron-dense junctions to support the statement in the text. It is hard to know what is meant by lateral expansion, especially when only showing a small portion of the cell membrane. The mild phenotypes are likely due to mild RNAi.
Response: We agree it is difficult to see the lateral membranes, so we added dotted lines to show them. Some of the separations seen are not so “mild”, especially for arp-2 RNAi. We agree RNAi does not fully remove these proteins, so yes, the mild phenotypes are due to mild RNAi, and yet the membranes are affected.
Discussion
Formatting issues.
Lines 284-286- says puzzle twice, edit to remove one.
Response: The second “puzzled” was replaced with “seemed incongruous”
Combine Lines 284-300 into one paragraph.
Line 293 – confusing as written. The text describes the apical localization of cadherin, but then basal localization is emphasized. In the figure, arrows point to apical enrichments. I recommend describing what was seen first (summary of results), then tell us how this 1) fits with prior results and models, and 2) doesn't. For example, apicolateral localization may be expected based on E-cadherin localization in other cell types, but its basal localization may be unexpected.
Response: The Reviewer raises an interesting point. We did this study in part because of surprising reports that Cadherin was not at all polarized in this tissue. We think our data here show this is not true, when C. elegans E-Cadherin/HMR-1 is examined using endogenously tagged CRISPR strains. However, the pattern of Cadherin/HMR-1 enrichment is quite complex. Besides the differences along the lateral cell-cell membrane, we were surprised to see a significant enrichment basally. The cells are not forming adherens junctions at the basal regions (by EM). We mentioning the basal enrichment since in our system the basal region of the cells is required for the intestine to interact with surrounding tissues.
Line 307 – when referring to the biosensor, states ‘widening of lateral regions’ – but this wasn’t measured with this probe.
Response: We changed this to, “we saw more diffuse signal in the lateral region in animals depleted of gex-3.”
Switching between gex-3 and WAVE complex – be consistent.
Response: We changed this to “WAVE component gex-3”
Line 310 “since these animals stills ….” Should be still. Response: Fixed.
Line 321-322 “branched actin is supports …” remove is. Response: Fixed.
Clarify the model. Is it that branched F-actin could control trafficking, which could maintain a balance of cadherin at the apicolateral region? Or is it by competing with linear filaments to balance their number? Or are they providing different forces at junctions? It isn’t clear what the role of branched F-actin is proposed to be, and it would help the reader if the authors could speculate.
Response: We believe it is through Cadherin trafficking. We are continuing further studies to test this model. This has been added to the Discussion, and possible roles of branched actin are introduced more clearly in the Introduction.
Reviewer 3 Report
Cordova-Burgos et al. propose to investigate the role of branched actin in polarized epithelia. To this aim, they develop tools allowing for high resolution imaging of C. elegans epithelia both during development and in the fully formed tissue and find that the actin nucleator promoting factor WAVE is required to correctly locate Cadherin at apicolateral membranes.
The study is overall well conducted and the CRISPR-based tools generated by the authors are definitely valuable tools for this and further studies. It is worth to mention that there are not so many studies that focus on how polarity is maintained, so I wish to congratulate with the authors for facing this understudied aspect of cell polarity.
I have only a few minor comments that can probably be addressed in the text:
- It is not clear to me whether the authors used a control RNAi (control HT115 E. coli like in their previous publication listed in Ref 5) or if they compare RNAi animals with fully wt animals. Would it be possible to specify this in the materials and methods or in the figure legends?
- In Figure 3C the authors show E-Cadherin/hmr-1 depletion by RNAi, but the fluorescent hmr-1 is still present. Is the hmr-1-mKate2 fluorescence intensity reduced upon RNAi treatment? If not, would a scrambled RNAi treatment result in the same phenotype?
- On the same line, is it possible that the defects observed in the beautiful electron microscopy pictures presented in Fig 4 could be due to the feeding protocol used for RNAi? What would be the effect on apical junction after feeding C. elegans with control E.coli?
- By using PIP2 probes to mark membranes, the authors find an apical enrichment of PIP2 (ratio apical to lateral of 2.5 - line 243 in the text). I think it would be of help for the readers to represent the data of Fig 3B as ratio apical to lateral also for gex-3 RNAi and hmr-1 RNAi.
Other minor comments:
- Images in Fig 2D refer to gex-2 RNAi, but in the text (line 213) there is a reference to gex-3 RNAi;
- Figure 3B legend is missing;
- hmr-1 is sometimes referred to as Cadherin and some other times as E-Cadherin. Can the text be homogenized?
- Scalebars are missing on microscopy images.
Author Response
Cordova-Burgos et al. propose to investigate the role of branched actin in polarized epithelia. To this aim, they develop tools allowing for high resolution imaging of C. elegans epithelia both during development and in the fully formed tissue and find that the actin nucleator promoting factor WAVE is required to correctly locate Cadherin at apicolateral membranes.
The study is overall well conducted and the CRISPR-based tools generated by the authors are definitely valuable tools for this and further studies. It is worth to mention that there are not so many studies that focus on how polarity is maintained, so I wish to congratulate with the authors for facing this understudied aspect of cell polarity.
I have only a few minor comments that can probably be addressed in the text:
- It is not clear to me whether the authors used a control RNAi (control HT115 E. coli like in their previous publication listed in Ref 5) or if they compare RNAi animals with fully wt animals. Would it be possible to specify this in the materials and methods or in the figure legends?
Response: We used control HT115 containing the empty vector, L4440. This has been added to the Methods.
- In Figure 3C the authors show E-Cadherin/hmr-1 depletion by RNAi, but the fluorescent hmr-1 is still present. Is the hmr-1-mKate2 fluorescence intensity reduced upon RNAi treatment? If not, would a scrambled RNAi treatment result in the same phenotype?
Response: We added to Methods, and added a Supplemental Figure to directly address this, showing how effective RNAi was at causing embryonic lethality, and reduction of HMR-1::GFP in the intestinal cells. Control “empty vector” RNAi (the same vector typically used for RNAi studies, L4440, but with no insert DNA) does not cause any of these phenotypes.
On the same line, is it possible that the defects observed in the beautiful electron microscopy pictures presented in Fig 4 could be due to the feeding protocol used for RNAi? What would be the effect on apical junction after feeding C. elegans with control E.coli?
Response: We have done these controls and they do not produce large membrane separations, or junctions that become so short and lose electron density. We showed the EM data to David Hall, the top EM expert for C. elegans, and he ruled out that EM artifacts could produce these changes.
By using PIP2 probes to mark membranes, the authors find an apical enrichment of PIP2 (ratio apical to lateral of 2.5 - line 243 in the text). I think it would be of help for the readers to represent the data of Fig 3B as ratio apical to lateral also for gex-3 RNAi and hmr-1 RNAi.
Response: We have added this ratio to Fig. 3B.
Other minor comments:
- Images in Fig 2D refer to gex-2 RNAi, but in the text (line 213) there is a reference to gex-3 RNAi;
- Figure 3B legend is missing;
- hmr-1 is sometimes referred to as Cadherin and some other times as E-Cadherin. Can the text be homogenized?
- Scalebars are missing on microscopy images.
Response: Thank you for noting these. All have been addressed.
Round 2
Reviewer 2 Report
I appreciate the time that the authors took to address all of my comments. This has been done appropriately, and I have no further concerns or comments. This is a nice study that will be of interest to a broad readership in cell and developmental biology.
Author Response
Thank you for your thoughtful review and comments.